# Health Communication in the Time of COVID-19 Pandemic: A Qualitative Analysis of Italian Advertisements

**DOI:** 10.3390/ijerph20054424

**Published:** 2023-03-01

**Authors:** Rosa Scardigno, Pasquale Musso, Paolo Giovanni Cicirelli, Francesca D’Errico

**Affiliations:** Department of Educational Sciences, Psychology, Communication, University of Bari Aldo Moro, 70122 Bari, Italy

**Keywords:** COVID-19, advertising, health communication, cultural narratives, elaboration likelihood model

## Abstract

In the climate of great uncertainty characterizing the COVID-19 pandemic, health communication played a significant role: several communicative strategies and channels were used to inform, educate and alert. Entropy-related risks were soon translated into the “infodemic”, a wide-spread phenomenon with psychosocial and cultural roots. Therefore, new challenges for public institutions occurred: public health communication, especially expressed through advertising and audiovisual spots, was engaged to offer key support in combatting the disease, mitigating its effects and supporting health and psychological wellbeing. This work aims to investigate how the Italian public institutions addressed those challenges by employing institutional spots. We tried to answer two main research questions: (a) in line with the literature concerning persuasive communication, what were the main variables that social advertising concerning health attitudes and behaviors relied on; and (b) how the different variables were combined to propose specific communicative pathways following both the different waves/phases of the COVID-19 pandemic and the elaboration likelihood model. To answer these questions, 34 Italian spots were analyzed by means of qualitative multimodal analysis (including scopes, major narratives themes, central and peripheral cues). The results enabled us to individuate different communicative pathways, oriented by inclusivity, functionality and contamination, in line with different rounds as well as with the holistic configurations of cultural narratives, central and peripheral cues.

## 1. Introduction

The onset of the COVID-19 pandemic between the end of 2019 and the beginning of 2020 represented a phenomenon strongly correlated to a climate of great health uncertainty. After China, Italy was the second largest country with the most immediate and widespread diffusion of the SARS-CoV-2 virus. These data characterized it as having a privileged point of view and, in some respects, being a model of the various processes involved in the pandemic event. In Italy, as in other countries, the expansion of the virus, combined with the initial inability to provide adequate solutions for the treatment of the disease, immediately forced the national public health authorities to think of and plan a series of measures that could curb the spread of disease and the blocking of hospital systems. The resulting list of measures was accompanied by a significant communicative effort to seek the general population’s compliance and enable the established political and health strategies to be effective.

Communication in the health field has therefore played an essential role in the management processes of the COVID-19 pandemic. However, it is well recognized that public health is a highly difficult field in which to intervene with massive communication campaigns to inform citizens about health risks and disease prevention [1]. Since the beginning of the pandemic, an overall sense of greater “entropy” and communicative disorder has spread [2], in line with gradually emerging information linked to the initial definition and naming of the disease, the hypotheses about its origins and the continuously updated practical management that the socio-political-health authorities put in place [3]. Furthermore, this scenario has been complicated by the vast world of misinformation/disinformation, referred to as the “infodemic” by the World Health Organization [4].

These processes and problems have manifested themselves in a climate previously characterized by a progressive erosion of trust in public institutions and by a general state of information crisis in the field of health and science [5]. Consequently, various leaders and institutions opted for a communication diversification which, through the use of different channels (traditional and digital), could allow greater visibility and reliability of communication on public health. During the most intense phases of the pandemic, research testifies to greater access to the television medium in the daily life of the population [6] and, more generally, the crucial role played by traditional media (for example, the radio in addition to television) in the process of building trust with audiences in different countries of the world [7,8]. At the same time, there has been a growing use of social media as a digital medium with a strategic role. For example, the Italian Ministry of Health used its official Facebook page to mitigate the spread of misinformation and to offer updates on the pandemic [5].

Through these channels, institutional communication, following the evolution of the pandemic, has made effective use of public advertising, primarily through audiovisual spots, intending to represent the fundamental support in countering the SARS-CoV-2 infection, mitigating the effects of the disease and, in general, support the health and psychological well-being of the population. The present work is specifically focused on this short and strategically planned communication and analyzes the entire Italian institutional advertising campaign, broadcast during the COVID-19 emergency, through a qualitative investigation of their main characteristics.

### 1.1. Advertising during COVID-19 

It is widely acknowledged that advertising acts as a “cultural operator” and through micro-stories of everyday life, “it deliberately turns abstract notions into specific situations by precisely delineating features, contexts, dialogues and social interactions” [9] (p. 3). Spots configure cultural imageries, stereotypes and tropes [10] and are meticulously designed to portray and channel particular messages, shared by the involved agents, using specific visual, sound and textual methods. Specifically, in the world of advertising, connections and emotional links with the target audiences are usually guaranteed through the production tactics involving identification [11] and empathetic connections [12]. These features remain even when public institutions promote advertising concerning both political issues [13,14] and humanitarian goals [15].

Taking into account the colossal scale of the social, economic and political events related to COVID-19, the attention of social scientists and scholars should be focused not only for the scrutiny of the events themselves but also on how they are narratively reported: cultural narratives generated by advertising can account for new forms of awareness and sensibilities [9]. During the COVID-19 health emergency, in line with modified lifestyles and consumption habits, advertising had to adapt to these changing communities, even capitalizing or adapting audiovisual spots. Especially in the first months of the pandemic, a change in content, language and images has been found, aimed at utilizing an emotional message over a product sponsorship [16]. Thus, audiovisual spots played an essential role: brands gained a new social function and advertising changed its traditional role to offering key support in improving resilience, alleviating stress and catalyzing health and psychological management [8]. Besides these already important features, institutional communication had to encourage healthy behaviors (such as social distancing, mask-wearing and vaccination) in a general effort to obtain individuals’ compliance [17]. Thus, the research for methods for promoting behavior change becomes more and more time-sensitive.

For at least the past 60 years, social psychology widely acknowledged the importance of persuasive communication in trying to change peoples’ attitudes and behaviors, and these studies generated a consensus regarding the role of attitudes in affecting actions as well as concerning the existence of several variables moderating attitude–behavior relationships [18]. These pathways were analyzed in line with several relevant public/private domains, such as tobacco use [18], sustainable holiday choices [19] and HIV prevention [20], among other things. Specifically referring to this last issue, an interesting metanalysis [20] has pointed out the several theoretical backgrounds implied in the explication of the relations between persuasive communication and changing actions, ranging from the health belief model [21] and the protection–motivation theory [22] to theories of reasoned action [23] and of planned behavior [24], and from the social-cognitive theory [25] to the information–motivation–behavioral skills model [26]. Therefore, several variables are invoked to explain and propose successful persuasive communications, including beliefs and emotions, perceived desirability and normative pressure, perceptions and behavioral intentions, knowledge and behavioral skills [20]. In addition, even the combination between self-benefits and social norms has been identified as appealing in persuasive communications dealing with sustainable practices [19].

An integrative framework to analyze how different mechanisms in different situations can impact persuasion is the elaboration likelihood model (ELM; [27]). This model emphasizes the importance of motivation and the ability to elaborate a message as critical factors affecting how deeply individuals ultimately elaborate it, thus defining a dual route of persuasive message processing, namely central and peripheral routes. Being defined based on full vs. reduced active engagement and the evaluation of the information by the recipients, the central route involves complex cues, requiring extensive cognitive efforts, such as argument quality, rational appeals and informational cues. In contrast, the peripheral route encompasses more implicit and superficial aspects of the message, thus implying a more heuristic process, such as source attractiveness and prestige, emotional appeals, visual and sound effects and so on [28]. In addition, this integrative model proposes “a link between the amount of elaboration people put into forming or changing an attitude and the strength of that attitude, with greater elaboration leading to greater strength” [17] (p. 327). In other words, since not all persuasive communications are equal, the choice and use of different cues will imply different elaboration levels, thus producing different strength outcomes and, finally, different possibilities that attitudes will guide behaviors. A recent study concerning COVID-19 vaccination showed that both central and peripheral routes influenced individually perceived informativeness and perceived persuasiveness, in turn affecting attitudes towards vaccination and the intention to obtain the vaccine [29].

Since public health communication and campaigns aim, especially during a sanitary emergency, to obtain a massive adhesion to specific attitudes and behaviors, persuasive messages should be disseminated in the most conducive way. Consequently, the types of cues and the related levels of elaboration are of significant interest to the comprehension and improvement of the persuasive processes.

### 1.2. The Current Study

This work aims to investigate how Italian public institutions focused on health communication by means of institutional spots during the different phases of the pandemic crisis. In this light, cultural issues and persuasive pathways were considered. Specifically, this work tried to answer the following research questions: (a) in line with the literature concerning persuasive communication, what were the main variables/factors that social advertising relied on when trying to affect health attitudes and behaviors, and (b) how the different variables were combined in order to propose specific communicative pathways following the different waves/phases of the COVID-19 pandemic. These research directions were explored in accordance with several variables, concerning: (i) the scopes of the spots; (ii) the major cultural narratives proposed by Italian institutional advertising; and (iii) in accordance with the elaboration likelihood model, the main types of central and peripheral cues.

## 2. Materials and Methods

### 2.1. Data

We collected and analyzed 34 Italian spots (coinciding with the whole institutional campaign from March 2020 to December 2021), which were broadcasted during the first four COVID-19 waves through the institutional national TV channels (RAI channels), social media (e.g., YouTube) and social networks (institutional official pages and profiles). The whole corpus (see Table 1) was available at the official Italian government website (https://www.governo.it/it/node, accessed on 1 September 2022).

### 2.2. Coding Scheme and Procedure

Spots were analyzed through qualitative multimodal content analysis. Based on the literature concerning institutional advertising during the pandemic and of the more general application of ELM to advertising [9,17,28,30], a codebook on an Excel sheet was created (with each line as an item and each column as a variable). Two coders had a training session and were well-instructed on the different variables included in the research project; after having independently co-coded 20% of the sample, a joint discussion on disagreements was carried out and certain operational definitions were refined, thus obtaining a satisfactory reliability. In some cases, e.g., cultural narratives, values and gestures could be codified on more than one option.

The coding activity was conducted in accordance with the following domains.

(a)*General (meta)data*. In this domain, a categorization of the scopes of advertising was proposed, through bottom-up and top-down processes, as it enabled us to frame the main functions of spots.(b)*Cultural narratives*. We considered this variable essential since, through the scenarios and social interactions offered by the spots’ micro-stories, the representations of reality were transformed into cultural references, supporting how reality may be perceived and explained [9].(c)*Central cues*. As the favorite way for an accurate cognitive elaboration, the presence of information, the reference to morality and values and the type of argumentation were investigated.(d)*Peripheral cues*. The features of testimonials, images and soundtracks were considered significant but activated less elaborate reflections.

These domains and variables were included in our analysis as they were able to explain the institutional and communicative efforts to activate both more general meaning-making processes and more contextualized pathways of content elaboration. The specific codifying variables are presented in Table 2.

## 3. Results

In line with the broadcast date/time and with the main objects proposed by the audiovisual spots, three phases of the Italian institutional campaign concerning health communication during COVID-19 were identified.

(1) *Facing lockdown*. Spots from 1 to 12 (broadcast from 11 February 2020 to 23 April 2020) were included in this round. The main scopes of these spots concerned hygiene rules for virus prevention, the presentation of supporting services to facilitate the sanitary emergency, the promotion of virtuous behaviors and emotional messages of union/solidarity.

As for the cultural narrative domain (see Figure 1), the most recurring narratives concerned:
-call for collective responsibility and mutual protection (five spots);-macro social changes (four spots) (telephonic and online services for retired people, doorstep pensions, online school activities and ads in opposition to violence against women);-resilience and overcoming challenges (three spots);-feeling of togetherness (three spots);-sense of community (three spots);-space and social atmosphere (two spots);-gratitude (one spot) addressed to “our heroes” (physicians, security forces and other workers from various productive sectors).

Looking at the central cue domain (see Figure 2), five spots (on 12) have no explicit informative claims and substantially coincide with union-solidarity messages. The others, concerning rational/informative issues, presented data just in one case (n. 11: “in Italy a woman is killed every three days […]. From 2000 to today, 3230 feminicides were committed: 1564 by the hands of their partner/ex-partner”). The moral domain is more other-oriented (eight spots) than self-oriented (four spots): whereas in the first case, messages concerning reciprocity and mutual protection were proposed, and in the second case, references to vulnerable groups—e.g., older people, women at risk—or individual calls for self-protective measures were found. As for the values, eleven spots stressed the importance of responsibility and social justice for vulnerable people, accompanied by the value of national security (five spots); and the value of fighting against a war/challenge is conveyed in four spots. The argument slightly favors a one-side perspective, i.e., it was preferred when practices, behaviors and services are proposed, whereas a two-sided argument was used in order to anticipate possible targets’ reactions (n. 3: “It is fair staying home, these are the rules, and now even if you are young, it’s time to comply with them”).

As for the peripheral cue domain (see Figure 3), five spots either contained no testimonials or did not have well-identifiable ones (e.g., people are quickly shown and have no voice). In six spots, testimonials were celebrities, mainly from TV, cinema and theatre (four spots) and the music world (two spots). A unique spot involving sportspersons included 10 top athletes from different sports. Testimonials were directly related to COVID-19 in a single video (spot no. 7), showing physicians, pharmacists and other professional figures. As for the gesture, about half of the spots present iconic realistic items (e.g., washing hands and keeping distance): they accompany and strengthen what is said and model the audience’s behaviors as a function. A similar function was achieved by deictic (two spots) and batonic gestures (one spot), whereas a stricter emotional activation was promoted by symbolic ones (three spots). Spot no. 2 is “bilingual”, as it also uses Italian Sign Language, having a broader inclusive aim. As for the audiovisual domain, five spots show some kinds of moving images (e.g., empty public spaces and classrooms, children’s pictures); otherwise, almost neutral images and scenarios were shown. As for the soundtracks, eight spots were classified as emotional, four of which utilized well-known motifs/songs.

(2) *Living with COVID-19*. Spots from 13 to 24 (broadcast from 5 May 2020 to 28 October 2020) were included in this round. Even if officially set during the first Italian lockdown, spots from 13 to 15 were included as well, since they appeared as “transitional” messages aimed to convey public recommendations for facing the last days of lockdown. The main scopes proposed by audiovisual spots concerned hygiene rules for virus prevention, supporting services for the health emergency and the promotion of virtuous behaviors.

As for the cultural narrative domain (see Figure 1), we found:-collective responsibility and mutual protection (nine times);-sense of community (two times);-having a resilient attitude (two times);-social spaces and atmosphere (two times);-macro social changes (one time) (high school qualification in person);-the possibility of coming back (one time).

The central cue domain (see Figure 2) was constructed as follows: firstly, all the audio videos contained informative cues, even if no data, graphs or percentages were proposed; second, half of them presented both self-oriented and other-oriented morality, with just two cases being self-oriented and three being other-oriented. Direct calls mostly utilized this mixed morality (e.g., “cover your mouth, noise and chin well”, spot no. 13), having a both a subjective and public impact (e.g., “a simple precaution aimed to protect both your health and the health of others”, spot no. 13). As for values, responsibility and social justice for vulnerable people again played an essential role (nine times), accompanied by national security (six times) and fighting a war/challenge (two times). A reference to the research on COVID-19 (spot no. 16) was also shown. The argumentation was somewhat divided between one- and both-sided. When the second type of argumentation was used, it seems to hint at the possible public resistance to healthy attitudes and practices, as in the following example: “to wear the mask is not easy as it seems” (spot no. 19).

As for the peripheral cue domain (see Figure 3), five spots have no testimonials, since they present stylized persons, a visual illustration of the COVID-19 virus or other images. The other videos show both ordinary persons (of different ages, roles and positions) and celebrities (in three spots). Celebrities come from the TV/theatre world; specifically, two were popular comedians. Spots involving these actors have a familiar atmosphere and present funny inserts (e.g., dialect words, mistakes in proposing a Latin adage). During this stage, no testimonial was directly related to the pandemic. As for gestures, five spots again represented iconic realistic gestures (e.g., washing hands, wearing the mask); in addition, both indexical (four times) and batonic gestures (three times) recurred. More generally, gesture appears intentionally marked to emphasize the advertising aims, which in this phase were similar to those of the first round. One spot also presented symbolic gestures, whereas spot no. 18 made use of realistic gestures with symbolic functions: opening the door, turning off the PC and taking off the mask all represent (prudently) regained freedom. As for the images, four spots presented emotional scenarios/activities (e.g., smiling, playing, empty spaces), whereas the most considerable part contained relatively neutral images (e.g., everyday places and activities). Just three spots had an emotional soundtrack, whereas the others were classified as neutral. However, no popular songs/motifs were used.

(3) *The vaccine challenge*. Spots from 25 to 34 (broadcast from 17 January 2021 to 29 December 2021) were included in this round. The scope of the spots concerned the vaccine awareness campaign and, just in one case, the supporting services for the health emergency (but again related to the vaccine facilitation).

As for the cultural narrative domain (see Figure 1), we observed the following:-collective responsibility and mutual protection (five times);-social spaces and atmosphere (eight times);-resilience and overcoming challenges (seven times);-togetherness (six times);-sense of community (four times);-the idea of coming back to past situations (three times);-macro social changes (two times), referring to the risks of vaccine-related fake news;-gratitude (one spot) addressed to members of the scientific field for their work with vaccines.

Looking at the central cue domain (see Figure 2), all the spots (except no. 25, which displayed explicit emotional elicitation) offered propositional content related to behavioral rules and vaccine application. However, data were provided in just two cases (e.g., “the vaccine reduces up to 90% the risk to going to the intensive care unit”, spot no. 32). As for morality, five spots were other-oriented, three were person-oriented and three displayed mixed morality. The value of responsibility and social justice for vulnerable people played again a central role (seven times), accompanied by the values of fighting a war/challenge (seven times) and national security (six times). More than in the other rounds, references to the research on COVID-19 also recurred (four times). Argumentation was widely two-sided (nine spots): the legitimacy of doubting (spot no. 25) and the ease of being taken in by a hoax (spot no. 29) led to the addressing of socially widespread worries (e.g., “[vaccine] have passed all the testing procedures concerning safety”; spot no. 32).

As for the peripheral cue domain (see Figure 3), spots frequently made use of mixed testimonials (both celebrities and common people, seven times), whereas either popular or unknown persons were present in two spots. Celebrities were mainly from TV/theatre (six spots), sport (five spots) and music (four spots). In addition, the protagonists of four videos were also associated with the COVID-19 pandemic (physicians and/or researchers). As for gestures, two features were found: (a) the presence of (again marked) mixed gestures, categorized in all types; (b) the occurrence of a specific repeated and dynamic gesture, represented by the iconic–symbolic “V” (made through the conjoint opening of the index and middle fingers), that is first placed on one’s arm (to represent the act of vaccination iconically) and, in a second moment, is held in front of the body (to represent the victory symbolically). This value–charge gesture clearly communicates the association between the vaccination campaign and the pandemic defeat. As for the audiovisual cues, just two spots had neutral images and only a single one had a neutral soundtrack; the prevalent emotional soundtracks coincide with popular songs/motifs (two popular Italian singers specifically created one song for this situation). Therefore, broad emotional audiovisual activation was provided.

## 4. Discussion

During the health emergency related to the COVID-19 pandemic, extraordinary interventions and measures were taken. Health communication had to be “resilient” to face the changing global community and meet citizens’ needs and expectations, trying to maintain responsible relationships with media and various strategic public institutions [5]. Institutional communication attempted to lessen collective uncertainty and promote cooperative attitudes and behaviors by emphasizing individual and social responsibility and trust [31]. As a largely shared challenge to persuade populations to adopt behavioral changes, it was essential to understand how these persuasive efforts were organized in a nation with a privileged view of the situation, namely Italy. The qualitative multimodal content analysis of the Italian institutional advertising campaign enabled us to propose the following insights.

In terms of the earliest and most critical phase, during lockdown, institutional advertising had to accompany the severe restrictions characterizing that period, facing uncertainties, informational needs and emotional turmoil. Two main directions were found in public spots. First, a wide variety of messages and cues, involving several cultural narratives and heterogeneous scopes, informative and conative aims and individual and collective values were evidenced. The well-blended narratives and the rather equally distributed central and peripheral cues, typical of this phase, appear oriented to emphasize the exceptionality of the situation. Second, the overall value of “inclusivity” was proposed, since: (i) scopes, narratives and contents were addressed to fragile populations (retired persons, at-risk women, students), as also testified by the use of Italian Sign Language; (ii) morality was mostly other-oriented and values were focused on responsibility and national security, thus emphasizing self-transcending and conservative orientations [30]. These features, together with one-sided argumentations and realistic gestures, supported the search for a widespread and popular elaboration of contents, involving multiple cultural narratives and not necessarily extensive cognitive efforts, to face the emergency pandemic period. On the one hand, this mixing and including style recalls some features of the protection–motivation theory [22], especially concerning the beliefs of personal susceptibility; on the other hand, it emphasizes the relevance of normative pressures, typical of more reasoned approaches [23].

The second phase, focusing on the need for co-existence with the pandemic, had a transitional nature: some cultural narratives and scopes typical of the initial stage (e.g., togetherness, union/solidarity, gratitude) were overcome in favor of more functional messages and cues. In addition, certain features typical of the third phase began to appear. The configuration of narratives represented a first timid impulse toward the idea of a “normal” life. The scopes, mainly oriented to present hygiene rules, supporting services, and virtuous behaviors; contents, dealing with simplified and repeated behaviors and practices; morality, both self- and other-oriented; and values, again emphasizing self-transcending and conservative orientations, concurred to promote a clear and univocal attitude toward health practices. The types of arguments, mixed between one-sided and two-sided but involving a larger number of issues, implied a more functional communicative style. In addition, the fewer recurring peripheral cues, mainly images and soundtracks, outline a more rational approach. In this context, testimonials were either missing or non-popular; when celebrities were used, three comic actors generated an atmosphere of familiarity and reassurance; therefore, they could be identifiable as non-biased testimonials [17]. The presence of marked gestures, which have a modelling and strengthening function, move this peripheral cue even closer to the central pathway. This stage appears to be more in line with a socio-cognitive approach [32]: since people are more likely to perform a behavior once they acquire the relevant knowledge and behavioral skills, persuasive communications should try to successfully model behavioral skills [20].

The third phase, essentially identified with the institutional pro-vaccine campaign, showed a more propositional aim and followed a new communicative scenario. The cultural narratives, again focused on responsibility and mutuality (representing the common thread of the whole campaign) were set in a changed context and appeared as future-oriented: the memory of social spaces and atmospheres, the emphasis on resilience and national security, the battle against the pandemic and the trust in science converged to create a general encouraging climate, promoting the opportunity to both come back and to overcome the challenges proposed by the pandemic. The main feature of the institutional campaign in this phase is “contamination”, which can be defined in the following forms: (1) the chronotope, since space and time, which were “suspended” in the previous stages, are proposed again, thus creating a connection between past practices, present care and future opportunities; (2) testimonials, with (i) both celebrity and non-celebrity figures appearing in the same spots and (ii) celebrities from all domains (TV, music and sport), as well as (iii) COVID-19-related non-celebrity figures; (3) central and peripheral pathways, involving values—not only highlighting self-transcendence and conservation but also openness and self-enhancement [30]—and contents and arguments (mostly both-sided), as well as images and soundtracks (broadly emotional and popular). In addition, a specific kind of contamination was proposed by gestures, again playing an essential role in institutional advertising, specifically in its ad hoc created dynamics and repeated configuration (from vaccine to victory). More generally, although this phase presented the highest number of peripheral cue occurrences (see Figure 3), hypothetically implying a reduced need for extensive cognitive activities, the presence of equally important central cues, such as informative references and two-sided argumentations, demonstrated the need to take into account the sensitivity and the variety of vaccine-related attitudes, making use of all the possible cues and persuasive pathways. Similarly to other stances, vaccine attitudes can be objects of ambivalence, one of the most promising moderators of the attitude–behavior link [18]. Therefore, focusing on the perceived desirability of the behaviors [23] and social pressure [29] can increase the engagement related to COVID-19 vaccines.

This work has some limitations, mainly related to the reduced and contextualized sample, even though it coincided with the Italian institutional advertising campaign. In addition, the variables, although they were selected from among the most representative in the recent international literature devoted to these matters and fit our sample well, were not exhaustive in terms of the central/peripheral cues in advertising. Most importantly, since pandemic communication was an important point for mitigating the crisis’ effects, an approach oriented to investigate the effectiveness of the applied communication and social advertising could fulfill the insights obtained from our results, mainly focused on the communicative features of the spots which the public institutions in Italy commissioned. This input represents an essential base for future investigation.

Nonetheless, to the best of our knowledge, this is the first work matching the analysis of cultural narratives with the ELM framework and applying the ELM model to institutional advertising concerning the health emergency of COVID-19. In addition, some emerging results—e.g., the types of arguments and the reference to values, the importance of gestures, and the role of testimonials—offer innovative implications for health communication and literacy and new inputs for audiovisual institutional communication.

## 5. Conclusions

Our study emphasized the importance of qualitative investigation as an opportunity to deepen the communicative efforts by institutions in the battle against the COVID-19 pandemic and the infodemic, in line with strategical advertising communication and against a broader socio-cultural background. Even if we propose a coding activity enabling us to compare the spots of each round referring to the incidences of cultural narratives and central and peripheral cues, we believe that the most interesting insights concern the way in which the included variables offer specific inputs and, at the same time, are set together as more holistic repertoires in different contexts. Therefore, similar narratives, as well as the preference for central/peripheric cues, can have different outcomes when they are differently blended, thus configuring communicative pathways regarding inclusivity (first round), functionality (second round) and contamination (third round).

In addition, this study offers significant applicative opportunities in promoting overall and specific awareness about health communication. In particular, the comprehension of the configuration of the cultural narratives and of the persuasive (central/peripheral) cues can (a) inform about the levels of elaboration and attitude creation, including epistemic self-defense skills and health literacy [33]; (b) improve effective communicative patterns in public communication and social advertising by institutional actors, also enhancing targeted and contextualized messages and trying to restore trust in institutions; (c) offer qualitative support for quantitative tools based on deep learning methods, as well as contesting the spread of misinformation and the “crisis” of discerning disinformation from accurate news [34].

## Figures and Tables

**Figure 1 ijerph-20-04424-f001:**
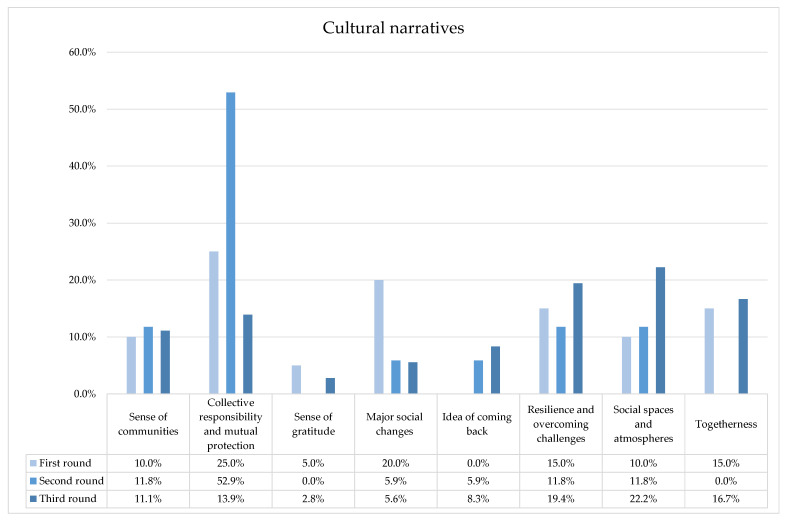
Percentage distribution of spots for each cultural narrative across the three phases of the Italian institutional health campaign during the COVID-19 pandemic.

**Figure 2 ijerph-20-04424-f002:**
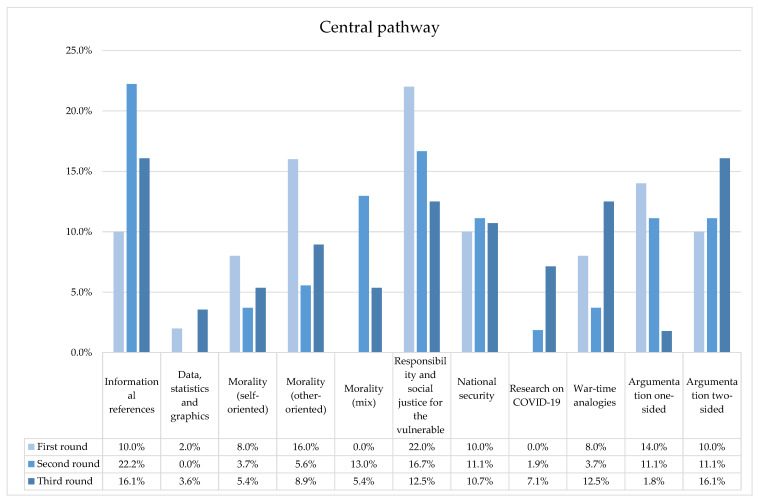
Percentage distribution of spots for each central cue across the three phases of the Italian institutional health campaign during COVID-19 pandemic.

**Figure 3 ijerph-20-04424-f003:**
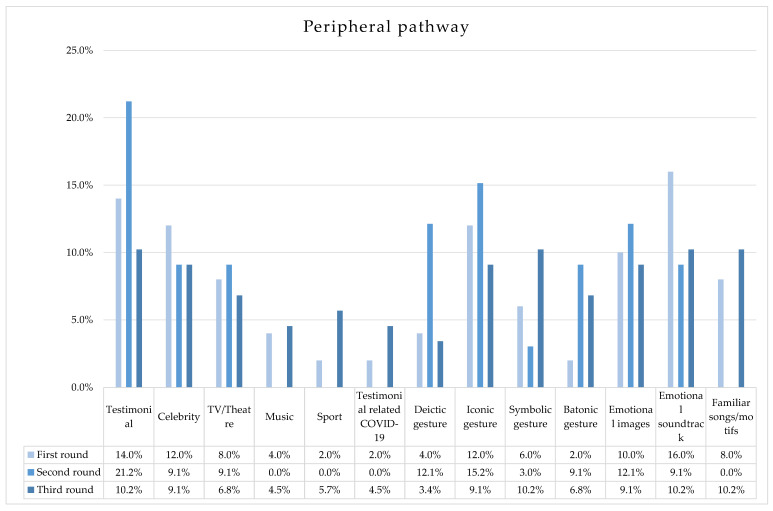
Percentage distribution of spots for each peripheral cue across the three rounds of the Italian institutional health campaign during COVID-19 pandemic.

**Table 1 ijerph-20-04424-t001:** The corpus of the analyzed audiovisual institutional spots.

Number	Spot Title	Publication Date	Main Scope
1	Protect yourself and the others—“Wash your hands well”	11 February 2020	Hygiene rules for virus prevention
2	Protect yourself and the others—“Let us help each other, together we can”	26 February 2020	Hygiene rules for virus prevention
3	#distantbutclose ^1^	11 March 2020	Union/solidarity
4	Stay at home—“Just do your job, spread the message among your contacts”	14 March 2020	Hygiene rules for virus prevention
5	INPS. Close even from afar.	19 March 2020	Supporting services for sanitary emergency
6	Do your part too. Stay at home.	23 March 2020	Promotion of virtuous behaviors
7	Thank you.	26 March 2020	Union/solidarity
8	I stay at home.	10 April 2020	Union/solidarity
9	Staying at home #schoooldoesnotstop ^1^	14 April 2020	Supporting services for sanitary emergency
10	Doorstep pensions	15 April 2020	Supporting services for sanitary emergency
11	Free, you can be.	15 April 2020	Union/solidarity
12	Everything is gonna be ok.	23 April 2020	Union/solidarity
13	To come back to smile together.	5 May 2020	Hygiene rules for virus prevention
14	It’s the distance that makes a real difference.	16 May 2020	Hygiene rules for virus prevention
15	Coronavirus can still touch us personally. It is up to us to stop it.	16 May 2020	Hygiene rules for virus prevention
16	Serological survey COVID-19	26 May 2020	Supporting services for sanitary emergency
17	App Immuni	11 June 2020	Supporting services for sanitary emergency
18	High school diploma 2020—#schooldoesnotstop ^1^	17 June 2020	Supporting services for sanitary emergency
19	Correct use of masks.	25 June 2020	Hygiene rules for virus prevention
20	Proper disposal of masks and gloves.	6 July 2020	Promotion of virtuous behaviors
21	App Immuni “the more we are, the better we keep”	31 August 2020	Supporting services for sanitary emergency
22	Download Immuni. Help yourself, your family and your country	7 October 2020	Supporting services for sanitary emergency
23	#iwearthemask ^1^	9 October 2020	Hygiene rules for virus prevention
24	Let us respect these #threesimplerules ^1^	28 October 2020	Hygiene rules for virus prevention
25	The hug room	17 January 2021	Awareness campaign for vaccine
26	Vaccines at the workplace. Together for a safe restart.	4 June 2021	Supporting services for sanitary emergency
27–31	Let us take back the taste of the future. (five spots)	21 June 2021	Awareness campaign for vaccine
32	Let’s do it for us.	15 November 2021	Awareness campaign for vaccine
33–34	Let’s do it for us. (two spots)	28–29 December 2021	Awareness campaign for vaccine

^1^ Some spot titles included a hashtag as a direct connection with social media platforms as well as because its use is an appealing feature for the target population.

**Table 2 ijerph-20-04424-t002:** The codebook variables.

Domain	Variable	Bibliographic Reference (If Any)	Coding Activity (When Predefined)
General and meta data	Spot title		
	Broadcast date		
	Url		
	Platform		
	Full text transcription		
	Main scopes of the spot		(1) Hygiene rules for virus prevention (2) Union/solidarity (3) Supporting services for sanitary emergency (4) Promotion of virtuous behaviors (5) Awareness campaign for vaccine
Cultural narratives	Main narratives	Grau-Rebollo, 2021 [9]	(1) Sense of community (2) Appeals to collective responsibility and mutual protection (3) Sense of gratitude (4) Representations of major social changes (e.g., remote working, hoaxes and intentional misinformation and so on) (5) Idea of coming back (6) Resilience and overcoming challenges (7) Social spaces and atmospheres (8) Togetherness (both literal and metaphorical)
Central cues	Informational references	Segev & Fernandes, 2022 [28]	(1) Yes (2) No
	Data, statistics and graphics	Segev & Fernandes, 2022 [28]	(1) Yes (2) No
	Morality	Susmann et al., 2022 [17]	(1) Self-focused (2) Other-focused
	Values	Wolf et al., 2020 [30]	(1) Responsibility and social justice for the vulnerable (2) National security (3) Research on COVID-19 (4) War-time analogies or achievement challenges
	Type of argumentation	Susmann et al., 2022 [17]	(1) One-sided (2) Two-sided
Peripheral cues	Testimonial	Segev & Fernandes, 2022 [28]	(1) Yes (2) No
	Celebrity	Segev & Fernandes, 2022 [28]	(1) Yes (2) No
	Celebrity domain		(1) TV/theatre/cinema (2) Music (3) Sport
	Testimonial related to COVID-19	Segev & Fernandes, 2022 [28]	(1) Yes (2) No
	Gesture acted by protagonists	Poggi, 2006 [14]	(1) Deictic ^1^ (2) Iconic ^1^ (3) Symbolic ^1^ (4) Batonic ^1^
	Emotional images	Segev & Fernandes, 2022 [22]	(1) Yes (2) No
	Emotional soundtrack	Segev & Fernandes, 2022 [22]	(1) Yes (2) No
	Familiar songs/motifs	Segev & Fernandes, 2022 [22]	(1) Yes (2) No

^1^ A brief description of each gesture is here proposed. (1) Deictic: to indicate something/someone by means of the index finger (or by the hand). (2) Iconic: to mimic the features/movements of something/someone. (3) Symbolic: to make use of a gesture that, in a specific culture, can be easily translated into words/sentences. (4) Batonic: to spell out and emphasize what is said by means of a top-down movement of the hands.

## Data Availability

Not applicable.

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
