# Peer review of "Health Communication in the Time of COVID-19 Pandemic: A Qualitative Analysis of Italian Advertisements"

_ijerph, 2023, doi:10.3390/ijerph20054424_

Round 1

Reviewer 1 Report

Comments

You have investigated an interesting question. However, I want to see how you can develop a research question which could have a dialogue with extant literature/theories and also enable you to address the practical issues.

In the introduction, you described the general background of the research project. The significance of the research was demonstrated, and the research context was also described ‘planned communication/ institutional advertising campaign, Spots, and their main characteristics”. In 1.1, you further elaborated the importance of studying these institutional advertising campaigns because of the importance of behavioral change in the pandemic. It was great! You continued to elaborate on the significance of achieving ‘A massive adhesion to specific attitudes and behaviors” by using advertising and by understanding “the types of cues and the related levels of elaboration”.  You mentioned ELM model but you did not really explain what ELM model is. I could see how your argument was going. Unfortunately, you just stopped here.

You possibly could research into the theories related to adhesion, theories of persuasion etc. Based on reviewing of the theories, you would develop a research question which would enable you to address the practical issues you described. You would also contribute to the extant literature. Based on a clear research question, you can further describe your research method and research process and then present your empirical research findings.

Author Response

REVIEWER'S COMMENTS

You have investigated an interesting question. However, I want to see how you can develop a research question which could have a dialogue with extant literature/theories and also enable you to address the practical issues.

In the introduction, you described the general background of the research project. The significance of the research was demonstrated, and the research context was also described ‘planned communication/ institutional advertising campaign, Spots, and their main characteristics”. In 1.1, you further elaborated the importance of studying these institutional advertising campaigns because of the importance of behavioral change in the pandemic. It was great! You continued to elaborate on the significance of achieving ‘A massive adhesion to specific attitudes and behaviors” by using advertising and by understanding “the types of cues and the related levels of elaboration”.  You mentioned ELM model but you did not really explain what ELM model is. I could see how your argument was going. Unfortunately, you just stopped here.

You possibly could research into the theories related to adhesion, theories of persuasion etc. Based on reviewing of the theories, you would develop a research question which would enable you to address the practical issues you described. You would also contribute to the extant literature. Based on a clear research question, you can further describe your research method and research process and then present your empirical research findings.

RESPONSE

Thanks for your enlightening review and for these holistic suggestions.

A wider overview of literature concerning persuasive communication and its effects has been proposed. This enabled us to propose more appropriate research questions, in dialogue with the proposed theories and, at their turn, to improve the other sections. Specifically, the research method has been improved by justifying the selected domains and variables; in the discussion, more explicit connections with the theoretical background were argued; practical issues were better specified. As a consequence, also bibliographic references have been enlarged involving both classical and recent items. As for ELM model, it has been deepened especially concerning its implications for this research.      

A linguistic review was also proposed.

Reviewer 2 Report

The authors chose a timely and insufficiently discussed topic when they scrutinized the health communication spots related to the COVID-19 pandemic, which were commissioned by public institutions in Italy. The analysis of the selected 34 spots was done in the system of a social psychological model (ELM). This seems very accurate, but the results and their interpretation are somewhat artificial. Pandemic communication was a weak point of fight against COVID-19 worldwide and proved to be a sensitive area. Perhaps that is why it would have been worthwhile to escape from the protective umbrella of the ELM and, for example, conduct guided interviews with one or more representative focus groups. This way, in my opinion, it would have been possible to obtain more information about the effectiveness of the applied communication and social advertising. Obviously, different messages are needed for each age group and type of settlement, but this is difficult to implement in a crisis situation. Personally, I would also have been interested in whether the spots were somehow able to counterbalance the tremendeous quantity of fake news spreading on the world wide web, in general, where and in what form these spots were spread (television, radio, social media, giant posters, etc.).

However, since, as far as I know, a similar communication has not yet been published, I recommend it for acceptance, despite the fact that the evaluation of health communication is narrowed down to cultural narratives. My proposals are intended to stimulate further exploratory work.

Author Response

REVIEWER'S COMMENTS

The authors chose a timely and insufficiently discussed topic when they scrutinized the health communication spots related to the COVID-19 pandemic, which were commissioned by public institutions in Italy. The analysis of the selected 34 spots was done in the system of a social psychological model (ELM). This seems very accurate, but the results and their interpretation are somewhat artificial. Pandemic communication was a weak point of fight against COVID-19 worldwide and proved to be a sensitive area. Perhaps that is why it would have been worthwhile to escape from the protective umbrella of the ELM and, for example, conduct guided interviews with one or more representative focus groups. This way, in my opinion, it would have been possible to obtain more information about the effectiveness of the applied communication and social advertising. Obviously, different messages are needed for each age group and type of settlement, but this is difficult to implement in a crisis situation. Personally, I would also have been interested in whether the spots were somehow able to counterbalance the tremendeous quantity of fake news spreading on the world wide web, in general, where and in what form these spots were spread (television, radio, social media, giant posters, etc.).

However, since, as far as I know, a similar communication has not yet been published, I recommend it for acceptance, despite the fact that the evaluation of health communication is narrowed down to cultural narratives. My proposals are intended to stimulate further exploratory work.

RESPONSE

Thank you very much for this review. We agree with your reflections concerning the aim of the research, consequently we added in the limitations some considerations about the need for the investigation of effectiveness of the advertising campaign. This perspective can sure represent one of the future steps of this research.

However, in this update version, we tried to better contextualize our work as an analytical proposal of the advertising choices, language and style, rather than on its effective impact.

A linguistic review was also proposed.